# Effect of Home Exercise Training in Patients with Nonspecific Low-Back Pain: A Systematic Review and Meta-Analysis

**DOI:** 10.3390/ijerph18168430

**Published:** 2021-08-10

**Authors:** Chloé Quentin, Reza Bagheri, Ukadike C. Ugbolue, Emmanuel Coudeyre, Carole Pélissier, Alexis Descatha, Thibault Menini, Jean-Baptiste Bouillon-Minois, Frédéric Dutheil

**Affiliations:** 1CHU Clermont-Ferrand, Department of Family Medicine, Faculty of Medicine, Université Clermont Auvergne, F-63000 Clermont-Ferrand, France; chloe.quentin@etu.uca.fr; 2Department of Exercise Physiology, University of Isfahan, Isfahan 81746-73441, Iran; will.fivb@yahoo.com; 3School of Health and Life Sciences, Institute for Clinical Exercise & Health Science, University of the West of Scotland, Glasgow G720LH, UK; u.ugbolue@uws.ac.uk; 4Physical Medicine and Rehabilitation, INRAE, UNH, CHU Clermont-Ferrand, Université Clermont Auvergne, F-63000 Clermont-Ferrand, France; ecoudeyre@chu-clermontferrand.fr; 5UMRESTTE, IFSTTAR, Université Lyon, 42100 Saint Etienne, France; carole.pelissier@chu-st-etienne.fr; 6UMRESTTE, IFSTTAR, Université St Etienne, 42100 Saint Etienne, France; 7Inserm, EHESP, Irset (Institut de Recherche en santé, Environnement et Travail), CAPTV-CDC, Université Angers, CHU Angers, F-49000 Angers, France; alexis.descatha@inserm.fr; 8Inserm, EHESP, Irset (Institut de Recherche en santé, Environnement et Travail), CAPTV-CDC, Université Rennes, F-49000 Angers, France; 9Department of Family Medicine, Faculty of Medicine, CHU Clermont-Ferrand, ACCePPT, Université Clermont Auvergne, F-63000 Clermont Ferrand, France; thibault.menini@uca.fr; 10Department of Emergency, CNRS, LaPSCo, Physiological and Psychosocial Stress, CHU Clermont-Ferrand, Université Clermont Auvergne, F-63000 Clermont-Ferrand, France; 11Occupational and Environmental Medicine, CNRS, LaPSCo, Physiological and Psychosocial Stress, CHU Clermont-Ferrand, WittyFit, Université Clermont Auvergne, F-63000 Clermont-Ferrand, France; fdutheil@chu-clermontferrand.fr

**Keywords:** musculoskeletal disorders, lumbalgia, physical activity, prevention, public health

## Abstract

Background: Exercise therapy is recommended to treat non-specific low back pain (LBP). Home-based exercises are promising way to mitigate the lack of availability of exercise centers. In this paper, we conducted a systemic review and meta-analysis on the effects of home-based exercise on pain and functional limitation in LBP. Method: PubMed, Cochrane, Embase and ScienceDirect were searched until April 20th, 2021. In order to be selected, studies needed to report the pain and functional limitation of patients before and after home-based exercise or after exercise both in a center and at-home. Random-effect meta-analyses and meta-regressions were conducted. Results: We included 33 studies and 9588 patients. We found that pain intensity decreased in the exclusive home exercise group (Effect size = −0.89. 95% CI −0.99 to −0.80) and in the group which conducted exercise both at-home and at another setting (−0.73. −0.86 to −0.59). Similarly, functional limitation also decreased in both groups (−0.75. −0.91 to −0.60, and −0.70, −0.92 to −0.48, respectively). Relaxation and postural exercise seemed to be ineffective in decreasing pain intensity, whereas trunk, pelvic or leg stretching decreased pain intensity. Yoga improved functional limitation. Supervised training was the most effective method to improve pain intensity. Insufficient data precluded robust conclusions around the duration and frequency of the sessions and program. Conclusion: Home-based exercise training improved pain intensity and functional limitation parameters in LBP.

## 1. Introduction

Low back pain (LBP) is a major public health issue [1,2], commonly described as pain and discomfort, localized below the costal margin and above the inferior gluteal folds, with or without leg pain [3]. Non-specific LBP is defined as LBP not attributed to a recognizable known specific pathology (e.g., inflammatory, tumoral or infectious process) [3]. Pain intensity and functional limitation are major factors in the prognosis of LBP [4]. Exercise therapy is recommended as first-line treatment [2,5,6]. However, the availability of centers for exercise therapy is lacking in the public health system [2,5]. Considering that the home is the most accessible setting [7], home-based exercise may be of particular interest in the management of LBP [7]. To facilitate the comparison of results between studies and to enable the pooling of data in this systematic review, an international multidisciplinary panel recommended, inter alia, pain intensity and functional limitation as core outcomes [3,8,9,10]. To our knowledge, to date no meta-analysis has assessed the effects of home-based exercise on pain intensity and functional limitation in LBP. At exercise centers some meta-analyses have suggested that a reduction in the risk of LBP could be achieved via various aerobic and resistance exercise training sessions, pilates and stabilization/motor control [11]. However, a European recommendation highlighted the absence of a clear consensus on the best exercise therapy [2]. For home-based exercise in LBP, data are scarce. Although data are lacking around the effectiveness of home-based exercise, as well as data regarding the optimal intensity, frequency and duration of exercise, supervised exercise seemed to produce the best outcomes in exercise centers [2,5]. Individual characteristics, such as age, sex, or education [3], may also influence responses to the home-based exercise program.

In light of this, we conducted a systematic review and a meta-analysis in order to assess the effect of home-based exercise on the pain intensity and functional limitations in LBP. The secondary objectives of this study were to assess the influence of the types and modalities of home-based exercise, and to investigate the putative influence of sociodemographic and characteristics of patients in the treatment of LBP.

## 2. Materials and Methods

### 2.1. Literature Search

We reviewed all studies reporting on the effect of home-based exercise training on nonspecific LBP (i.e., LBP not consecutive to a specific pathology such as inflammatory, tumoral or infectious process) [3]. Animal studies were excluded. The PubMed, Cochrane Library, Embase and ScienceDirect databases were searched until 20 April 2021, with the following keywords: low back pain AND (exercise OR physical) AND home (details of the specific search strategy used within each database are available in Appendix A). The search was not limited to specific years and no language restrictions were applied. To be included, articles were required to simultaneously meet the five following inclusion criteria: (1) randomized controlled trials (RCTs); (2) population ≥ 16 years old; (3) with non-specific LBP (chronic or not); (4) evaluation of at least one of our main clinically relevant outcome (i.e., pain intensity or functional limitation); and (5) studies including home-based exercise therapy. Home exercise programs are defined as a series of exercises that patients complete at home for therapeutic gains or to improve physical capacity. Home exercises are designed to be practical, accessible and feasible so that patients can maximize efforts. We excluded those studies which assessed patients with specific LBP (i.e., caused by a specific cause such as pregnancy or pathological entities). Conference papers, congress, and seminars were excluded. In addition, the reference lists of all publications meeting the inclusion criteria were manually searched in order to identify any further studies not found through the electronic search. Ancestry searches were also completed on previous reviews to locate other potentially eligible primary studies. Two authors (Chloé Quentin and Reza Bagheri) conducted the literature searches, reviewed the abstracts and, based on the selection criteria, determined the suitability of the articles for inclusion, and extracted the data. When necessary, disagreements were resolved with the inclusion of a third author (Frédéric Dutheil).

### 2.2. Data Extraction

The data collected included: (1) characteristics of the study, including the first author’s name, publication year, country and continent, study design, outcomes of included articles, and number of participants; (2) characteristics of individuals, such as the mean age, sex (percentage of males), weight, height, and body mass index, percentage of smokers and regularly physical active individuals, education and marital status, and duration of complaints; (3) characteristics of the intervention, such as whether the intervention was supervised (totally supervised/partially supervised/not supervised), standardized or individualized (partially or fully), the type of intervention (education, aerobic exercise, stretching, strengthening, relaxation, postural exercise, yoga, other exercises), the frequency and duration of sessions, the duration of the program, and the location of training (home or other setting); and (4) characteristics of our main outcomes, such as the type of assessment of pain intensity and functional limitation, and measures (mean and standard deviation) before and after the training.

### 2.3. Quality of Assessment

We used the Scottish Intercollegiate Guidelines Network (SIGN) criteria designed for randomized clinical trials to check the quality of included articles. The checklist consists of 10 items. We gave a general quality score for each included study based on main causes of bias. We used 4 possibilities for scoring each item (yes, no, can’t say or not applicable) [12].

### 2.4. Statistical Considerations

We conducted meta–analyses on the effect of LBP exercise on pain intensity and functional limitation. P-values less than 0.05 were considered statistically significant. For the statistical analysis, we used Stata software (version 16, StataCorp, College Station, TX, USA) [13,14,15]. The main characteristics were synthetized for each study population and reported as a mean ± standard deviation (SD) for continuous variables and number (%) of the categorical variables. First, we conducted random-effect meta-analyses (using the DerSimonian and Laird approach [16,17]) on the effect of home-based exercise for LBP, by comparing levels of pain intensity or functional limitation after the training program versus baseline levels (i.e., before exercise). The results were expressed as effect sizes (ES, standardized mean differences—SMD). ES is a unitless measure centered at zero if pain intensity or functional limitation did not differ between after and before the training program. A negative ES denoted an improvement in the pain intensity or functional limitation of the patient (i.e., decreased levels of pain intensity or functional limitation after exercise compared to before). An ES of −0.8 reflects a large effect, −0.5 a moderate effect, and −0.2 a small effect. Following this, we conducted meta-analyses stratified on: (1) the location of the training program (exclusively home, or home plus another setting); (2) characteristics of intervention, whether it was supervised (totally supervised/partially supervised) or not, and standardized or individualized (partially or fully). We computed the aforementioned meta-analysis using all the measurement time. To verify the strength of our results, we computed sensitivity analyses using only the median time of follow-up and then using only the last time of follow-up. We evaluated heterogeneity in the study results by examining forest plots, confidence intervals (CI) and I-squared (I²). I² values are a common metric used to measure heterogeneity between studies and are easily interpretable. I² values range from 0 to 100%, and are considered low at <25%, modest at 25–50%, and high at >50% [18]. For example, a significant heterogeneity could be linked to the characteristics of the studies, such as sociodemographic, or the characteristics of the intervention. We searched for potential publication bias using funnel plots of all the aforementioned meta-analyses, in order to conduct further sensitivity analyses by excluding studies that were not evenly distributed around the base of the funnel. When possible (where there was a sufficient sample size), meta–regressions were proposed in order to study the associations between changes in pain intensity or functional limitation, and clinically relevant parameters such as sociodemographic (age, sex, body mass index, etc.), and the characteristics of the intervention (e.g., type of exercise, supervised or not, standardized or individualized, frequency and duration of sessions, and duration of programs). The results were expressed as regression coefficients and 95% CI.

## 3. Results

An initial search produced 24,699 possible articles. The removal of duplicates and use of the selection criteria reduced the number of articles reporting the effect of home-based exercise on LBP patients to 33 articles [5,19,20,21,22,23,24,25,26,27,28,29,30,31,32,33,34,35,36,37,38,39,40,41,42,43,44,45,46,47,48,49] (Figure 1). All included articles were written in English. The main characteristics of the studies are described in Table 1.

### 3.1. Quality of Assessment

Overall, the methodological quality of the included studies was good, with an average score of 75% for items meeting the criteria of the SIGN checklist, ranging from 40% [5,44] to from 90% [32,42,50]. All studies failed to include a blind assessment. All studies reported achieving ethical approval (Figure 2).

### 3.2. Study Designs of Included Articles

The included studies were published between 1998 [28] and 2020 [51] and conducted in various geographic locations, with all continents represented (Europe [19,25,27,28,30,32,34,36,37,38,39,48,51], North America [22,26,31,35,42,50], South America [5,29,44], Asia [23,24,40,41,43], Africa [20], and Oceania [35,45,46,49]. All the studies were RCT. Thirteen studies were monocentric [5,19,20,21,22,23,24,29,30,31,33,42] and eleven were multicentric [25,27,32,35,37,38,40,41,43,50,51]. Studies were single-blind [19,23,24,25,28,29,30,32,33,35] or not blind randomized trials [5,22,27,47,51].

### 3.3. Inclusion and Exclusion Criteria of Included Studies

All the included studies included adults, except one study that also included participants over 16 years old [39]. Globally, participants were recruited using electronic, newspaper or local advertisements [23,25,26,33,35,42,44,45,46,47,49,50] or from consultations with specialists or general practitioners [5,19,20,25,40,41,49]. Some studies included specific populations, such as healthcare workers [25,27,51], sedentary older adults [30,42], poultry industry slaughterers [21], only women [19], or individuals from a rural community [33].

### 3.4. Population

The sample sizes of the studies ranged from 22 [23] to 385 participants [39]. We included a total of 9,588 LBP patients. The ages of the participants were reported in all except two of the studies [5,44]. The mean age of LBP patients undertaking home-based exercise training was 49.3 years (95% CI 45.5 to 52.9), with ages ranging from 32.6 ± 11.5 [43] to 74.7 ± 6.0 [42] years old. Gender was not reported in 14 of the selected studies [5,21,22,24,34,38,40,43,44,46,47,50]. The mean proportion of men was 18% (95% CI 0.15 to 0.20), with the proportion of men in the studies ranging from 0 [19] to 53% [42]. The BMI of participants was reported in 16 of the selected studies. The mean BMI was 29.5 kg/m² (95% CI 28.3 to 26.9), with BMIs ranging from 21.5 ± 2.7 [33] to 32.7 ± 7.4 [50]. Other parameters were seldomly reported. The education status of participants was reported in eight studies [27,29,31,33,34,42,49], but degrees were not expressed in the same way across most of the studies. Smoking status was only reported in six studies [25,28,29,42,43,49], with a mean proportion of smokers of 4% (95% CI 2 to 5%), ranging from 0% [47] to 11% [29]. Only three studies mentioned leisure physical activity [20,29,43], with a mean proportion of regularly active patients of 10% (95% CI 6 to 15%), and proportions varying from 3 [43] to 18% [29].

### 3.5. Intervention: Characteristics of Exercise

#### 3.5.1. Type of Exercise

Nearly all of the selected studies (26 studies i.e., 79%) used strength-based exercises, mostly combined with other exercise [5,19,20,22,23,24,25,27,29,30,33,34,35,36,37,38,40,41,42,44,45,46,48,51]. Only two of the studies did not [26,32]. Education was included in sixteen of the studies [22,23,25,27,28,29,30,32,33,34,35,36,40,41,42,47,50,51], stretching in twenty-three studies [5,19,20,21,23,24,25,26,27,28,29,30,31,32,33,34,35,36,37,40,41,42,44,45,46,47,48,49,51], aerobic exercise in thirteen studies [5,22,24,27,28,30,32,33,34,35,36,42,49,50], postural exercise in eight studies [21,25,27,28,29,44,45,47], relaxation in eight studies [21,27,28,30,34,40,43,50], and yoga in four studies [26,29,30,52]. Other exercises were only occasionally reported, including Thai self-massage [23], spinal manipulative therapy [22], stress-controlling techniques and a behavioral approach [27], Jyoti meditation [48], breathing [31,50], and ball games associated with body awareness and circle training [34].

#### 3.5.2. Duration of Intervention

The duration of the invention was reported in all of the selected studies, with the programs lasting an average of 11.4 weeks, and varying from 2 weeks [43] to 2 years [25].

#### 3.5.3. Frequency and Duration of Sessions

On average, the studies reported 4.8 sessions per week—ranging from 1 [42,51] to 14 [26] sessions per week—with each session lasting 63 min—ranging from 9 [43] to 240 min [29]. The frequency of sessions was not reported in four of selected studies [30,35,39,45], and the duration of sessions was not reported in in thirteen studies [5,23,28,30,34,35,36,37,38,39,41,46,47].

#### 3.5.4. Standardization

Standardization was not reported in seven of the selected studies [23,30,35,39,41,47,51]. The exercises were standardized in twenty of the studies [5,19,21,25,28,29,31,32,33,34,35,36,37,40,43,45,46,48,49,50], partially standardized in six of the studies [27,29,38,42,44,47], and individualized in five of the studies [20,24,26,27,45].

#### 3.5.5. Supervision

Supervision was reported in all except two of the studies [23,34]. Exercises were fully supervised in eight studies [19,20,33,39,44,45,47,51], partially supervised in twenty-one studies [5,20,21,25,27,28,29,30,31,32,33,34,36,37,40,41,42,44,47,48,50], and not supervised in ten studies [19,28,35,36,42,43,46,48,49,51].

#### 3.5.6. Location

The location was reported in all of the studies. Exercises took place exclusively at home in twenty-one studies [5,19,21,22,23,26,28,30,35,36,37,38,41,42,43,44,46,47,48,49,51] and both at home and another setting (workplace, health center or training center) in eighteen studies [5,20,21,25,27,28,29,31,32,33,34,36,40,42,45,47,48,50].

### 3.6. Outcomes-Pain Intensity and Functional Limitations

Before and after the physical exercise program, the pain intensity was assessed in 27 studies [5,19,20,21,22,23,24,25,26,27,29,31,32,33,34,35,36,37,40,41,42,43,44,45,46,47,48,49,50] and functional limitations in 28 studies [5,19,21,22,23,24,25,26,27,28,29,30,31,32,33,34,35,37,38,39,40,42,44,46,47,48,49,50,51]. Participants’ pain intensity was evaluated in most studies using a visual analogue scale [19,20,21,23,25,26,29,32,35,36,41,43,44,48], or a numeric pain rating scale [5,22,24,33,34,42,45,46,47,49,50]. Other pain assessments were made using the West Haven-Yale Multidimensional Pain Inventory [27], the Brief Pain Inventory [31], the Borg CR-10 scale [37], and the Defense and Veterans Pain Rating Scale [40]. Functional disability was evaluated in most of the studies using the Roland-Morris Disability [5,19,21,22,29,30,31,36,39,40,42,47,48,49,50,51], the Oswestry Disability Index [23,32,33,34,35,37,38] or its modified version [24,28,46], except in three studies that used the French version of the Quebec back pain disability scale [25], the Short Form Health Survey physical component scale [27], and the Quebec Back Pain Disability Scale Questionnaire [44]. Assessments of the outcomes were made at a median time of three months after the beginning of the exercise program, ranging from half a month [43] to five years [37]. Some studies reported outcomes at different times during the exercise protocol [5,21,22,25,27,28,29,31,36,37,40,42,43,45,47,48,51].

### 3.7. Meta-Analysis on the Effect of Home-Based Exercise

Overall, home-based exercise training decreased pain intensity (effect size = −0.89, 95% CI −0.99 to −0.80) and decreased functional limitation (−0.73%, −0.86 to −0.59) for participants, regardless of an exclusive at-home location or not. Pain intensity decreased in a similar proportion between an exclusive at-home setting (−0.97, −1.14 to −0.79) and a combination of exercises at home and in another setting (−0.89, −0.96 to −0.74). Similarly, functional limitation also improved in both settings (−0.69, −0.93 to −0.46; and −0.93, −1.34 to −0.52, respectively) (Figure 3 and Figure 4, and Appendix B).

### 3.8. Stratification by Characteristics of Training

Stratification by the supervision of training demonstrated a decrease in pain intensity and an improvement of functional limitation regardless of the characteristics of the training. For pain intensity, a totally supervised training seemed to be the most effective in terms of decreasing the pain intensity (effect size = −1.19, 95% CI −1.31 to −1.06; versus −0.71, −0.82 to −1.06 for partially supervised training and −0.93; −1.1 to −0.77 for unsupervised training). For functional limitation, improvement seemed similar regardless of the level of supervision of training (−0.76, −0.88 to −0.65 for totally supervised training; −0.76; −1.0 to −0.52 for partially supervised training, and −0.60, −0.71 to −0.48 for unsupervised training). Concerning the standardization of training, both for pain intensity and functional limitation, a standardized protocol seemed to produce the greatest benefits (−0.95, −1.31 to −1.08 for pain intensity, and −0.96, −1.16 to −0.76 for functional limitation), whereas a partially individualized program had the lowest benefits (−0.67, −0.8 to −0.54 for pain intensity and −0.45, −0.64 to −0.26 for functional limitation). An individualized program demonstrated very wide confidence intervals for pain intensity (−1.00, −1.42 to −0.57) and was not significant for functional limitation (−0.32; −1.06 to 0.43) (Figure 4).

### 3.9. Metaregressions

There was no difference in the improvement of pain intensity and functional limitation depending on the setting (exclusive home-based exercise vs. home-based exercise combined with exercises in a center, *p* = 0.66). Totally supervised exercise produced better benefits regarding pain intensity when compared with partially individualized (coefficient −0.50, 95% CI −0.74 to −0.25) or not supervised (−0.28, −0.58 to −0.01) programs, with no influence on functional limitation. Standardized protocols had better benefits when compared to partially individualized training in relation to both pain intensity (−0.29, −0.55 to −0.03) and functional limitation (−0.51, −0.84 to −0.18). Despite most studies mixing different types of exercise, metaregressions demonstrated an improvement in pain intensity for pelvic (−0.63, −0.92 to −0.30), leg (−0.27, −0.52 to −0.02) and trunk (−0.36, −0.65 to −0.07) stretching. In contrast, pain intensity seemed to be exacerbated by relaxation (0.38, 0.12 to 0.65). Postural exercises seemed to have a deleterious effect on both pain intensity (0.35, 0.11 to 0.59) and functional limitation (0.24, 0.00 to 0.68). Yoga improved functional limitation (−0.94, −1.67 to −0.2). The volume of training (the frequency and duration of sessions, and duration of programs) was not associated with an improvement in pain intensity or functional limitation. A longer follow-up was associated with a decrease in pain intensity (effect size = −0.03. 95% CI −0.05 to −0.01) and functional limitation (−0.06; −0.10 to −0.02). Men were more likely to improve pain intensity (−0.16, −0.23 to −0.08, per 10% men) and functional limitation (−0.23, −0.30 to −0.16, per 10%men) to a greater extent than women. The training program was less effective for decreasing pain intensity in people with a higher body mass (1.01, 0.19 to 1.85, per 10 kg·m^2^). Other parameters, such as age, education status, smoking, or duration of symptoms, were not associated with decreased pain intensity and functional limitation (Figure 5).

### 3.10. Sensitivity Analyses

We computed sensitivity analyses using only the median time of follow-up (i.e., three months (Appendix C for meta-analysis and Appendix D for metaregressions)), and the last time of follow-up (Appendix E and Appendix F). All the results were similar. Because of the wide heterogeneity of the selected studies (all I-squared are >80%), we failed to reperform all the aforementioned meta–analyses after the exclusion of studies that were not evenly distributed around the base of the funnel (Appendix G).

## 4. Discussion

The main findings of this research were that home-based exercise training improved pain intensity and functional limitation in LBP patients, regardless the modality of exercises. Supervised training and standardized training improved pain intensity to the greatest extent, independently of the influence of the duration and frequency of the training. Training was less beneficial for women and for patients with a high body mass index.

### 4.1. The Benefits of Home Exercise Training on LBP Patients

This study is the first systematic review and meta-analysis of studies investigating the effectiveness of home exercise programs on pain and functional limitation in patients with LBP. Structured center-based programs have the advantage that the amount and quality of the training can be controlled and supervised, but these programs are expensive, limiting their implementation possibilities [7]. Moreover, many adults experience barriers to attending such programs, including a lack of affinity with the culture of fitness centers [7]. Home-based exercises are especially valuable because they require fewer resources [52] and less time from health institutions and health practitioners [7]. Our meta-analysis showed strong evidence that physical exercise training can take place at home to improve LBP, even though we found no studies comparing the same training program between home and another setting. Studies comparing home-based exercise to a control group without exercise [25,31,38,43] showed an improvement in pain intensity and functional limitation. Thus, home-based exercise training could be a cost-effective intervention in the treatment of LBP. If multiple short bouts of moderate-intensity physical exercise produce significant training effects [53], learning to integrate physical activity into daily life can become a main goal in the treatment of LBP. Moreover, home-based exercise also improves other comorbidities, such as knee osteoarthritis [54], obesity [55], depression [56], gait speed in people with Parkinson’s disease [57], chronic obstructive pulmonary disease [58] and reduces the risk of cardiovascular mortality [59].

### 4.2. Which Type of Exercise Training?

There is overwhelming evidence that regular physical activity is associated with reduced LBP [2,3,4,6,52,60]. The most appropriate exercise intervention is still unknown. Opinions differ over the optimal exercise modalities used to treat LBP. The “active ingredient” of exercise programs is largely unknown, although various exercise options are available [2,60,61]. Considerably more research is required in order to allow for the development and promotion of a wider variety of low cost, but effective exercise programs [2,3]. Our metaregression demonstrated the benefits of pelvic, leg and trunk stretching, in line with the literature [53]. However, drawing any firm conclusions on the best type of exercise is impossible because most studies integrated strength and aerobic training, precluding further analysis (e.g., there was a lack of reference groups that omitted either strength or aerobic training). The predominance of strength in the selected studies is due to the high level of proof of its efficacy in the treatment of LBP [11,53], whereas the benefits aerobic exercises are more under debate [3]. Easily-performed exercises produced noticeable benefits and supported adherence to home-based exercise programs [62]. While aerobic training was easily achievable at home [59], strength training may require more supervision, at least at the beginning [63]. However, strength training is still achievable at home in a wide range of pathologies [52,57]. Despite conflicting results in the literature [2,60] relaxation and postural exercise seemed ineffective in reducing LBP, as well as education alone [60,64]. Similarly to center-based exercise [60], we found that yoga improved functional limitation, as previous studies also showed. Yoga usually combines a wide variety of exercises channeled towards improving strength and flexibility [31], which may explain its positive effects on reducing LBP [3]. Finally, considering the high impact of long-term adherence to exercise [62], the appropriateness of exercise programs may be best determined by the preferences of both the patient [6] and therapist [2].

### 4.3. Supervision, Standardization, Frequency, and Duration of Exercise Training

Several authors claimed that supervised exercise therapy had proven to be effective in reducing pain and improving functional performance in the treatment of patients with non-specific LBP [2,5,6,29,60] whereas others showed that supervision did not significantly influence final outcomes [5,22,42] but did enhance participants’ satisfaction with care [42]. We showed that the best improvements in LBP were achieved through supervised exercise training. It is important to note that all the home-based exercise programs were prescribed by a physiotherapist or health professional with a degree-level qualification in exercise prescription. The majority of the home-based exercise programs in our review incorporated partial supervision [5,20,21,25,27,28,29,30,31,32,33,34,36,37,40,41,42,44,47,48,50], (i.e., the use of a variety of methods, including home visits by the therapist, occasional group-based sessions at a center or telephones calls). Some publications suggested that external sources of reinforcement, like monitoring, may serve to influence physical behavior [65]. In addition, applying supervised home-based exercise is possible to achieve in many ways, helping to optimize effectiveness of the training [66]. Importantly, our results were in favor of standardized exercise compared to individualized exercise, which may be discordant with the literature based on training in centers [2,26]. This may be explained by the fact that easily-performed standardized exercises can promote a better adherence [62], and could be more in line with home exercise, whereas individualized exercise may be more in line with practice in a center. Lastly, we failed to demonstrate an influence of the volume of exercise on reducing LBP, despite a strong dose-response relationship between physical activity and its overall benefits [2,3]. The absence of such a significant influence on our study may be due to the wide variety of exercise interventions available, and the inconsistency of the intensity and duration of exercise [3].

### 4.4. Predictors of Pain Intensity Improvements

Although no gender differences were found in relation to pain improvement after exercise in most publication [53,60], our study found strong evidence that males with LBP benefited the most from exercise training. Even if the included studies did not report on observance of exercise, women may lack the time to engage in a daily routine of training [67,68]. Fractionalization of an exercise bout into multiple bouts spread across the day may produce greater benefits and allow for greater adherence [69]. Interestingly, some studies also reported a higher prevalence of LBP in women [1,70]. In our meta regressions, age was not associated with pain improvement. This suggests that home-based exercise, even late in life, can be effective [71]. We also demonstrated that the benefits of exercise were less effective in individuals with a higher body mass index, in line with the literature [53]. Furthermore, individuals using medication, those with no heavy physical demands at work and individual recovery expectations are important parameters influencing the prognostics of LBP [1,4,53], although this was seldomly mentioned in the studies included in our review.

### 4.5. Limitations

Our study has some limitations. All studies were randomized and patients were not blinded to the interventions. Several biases could have been introduced via the literature search and selection procedure. We conducted the meta-analyses on only published articles, therefore they were theoretically exposed to publication bias. Meta-analyses also inherit the limitations of the individual studies of which they are composed. The availability of some individual characteristics limits the ability to assess all potential treatment effect. Only 11 studies [22,23,24,26,30,35,39,43,46,49,51] had groups exercising only at home. Hence, comparisons between the efficacy of home-based training versus training in a center cannot reflect a high level of proof—even if we only included randomized trials. Similarly, the lack of studies using a control group without exercise precluded further comparisons. Another major limitation of our meta-analysis is the lack of data on physical activity levels, as well as on medications used. Additionally, the heterogeneity between the study protocols and evaluation may have impacted the results. Some short time-frames (two weeks [43]) may also have been too short for a therapeutic effect. Some studies included targeted population [21,25,27,30,31,33,43,51], however the large sample size of over 10,000 individuals of all ages and categories promotes the generalizability of our results. Even if the weight of studies requires careful thought, because some studies had several measurement interval times and training duration, sensitivity analyses based on median or last time of follow-up time demonstrated similar results. Moreover, our method had the advantage of avoiding selection bias [72].

## 5. Conclusions

From the literature, it is concluded that home training can be successful if the training is done at home with friends from a community group taking part. Home-based exercise training improved pain intensity and functional limitation parameters in participants experiencing LBP. Supervised training and a standardized program seemed beneficial, although insufficient data precluded drawing any robust conclusions around the duration and frequency of sessions. Further dedicated randomized controlled trials in which information about the type and characteristics of home-based exercise are included are warranted.

## Figures and Tables

**Figure 1 ijerph-18-08430-f001:**
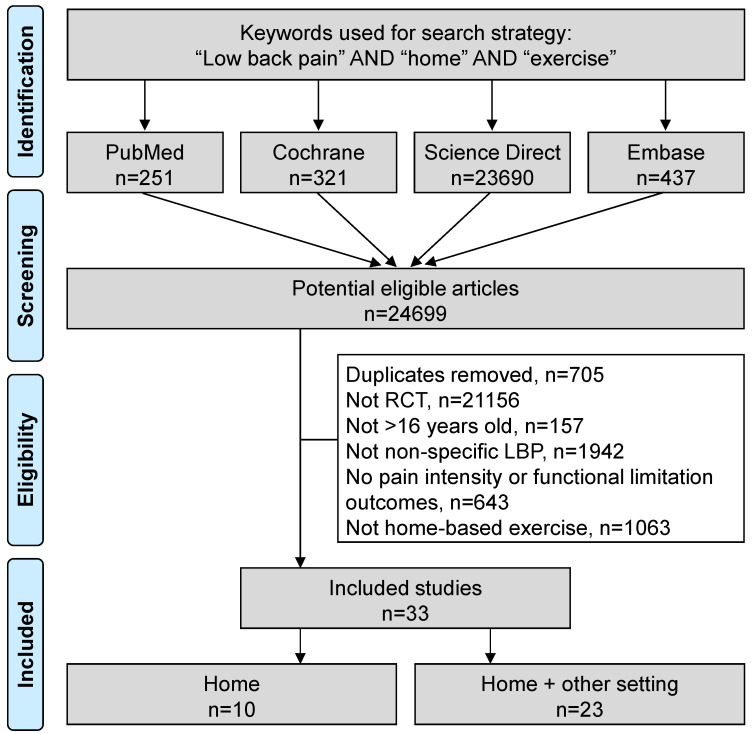
Flow chart.

**Figure 2 ijerph-18-08430-f002:**
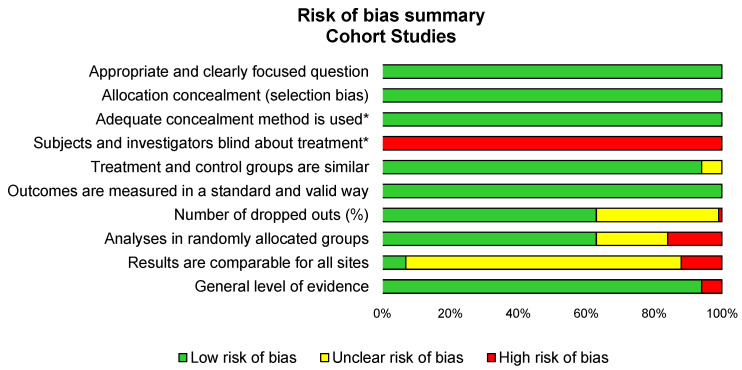
Summary of methodological qualities of included studies using the SIGN checklist.

**Figure 3 ijerph-18-08430-f003:**
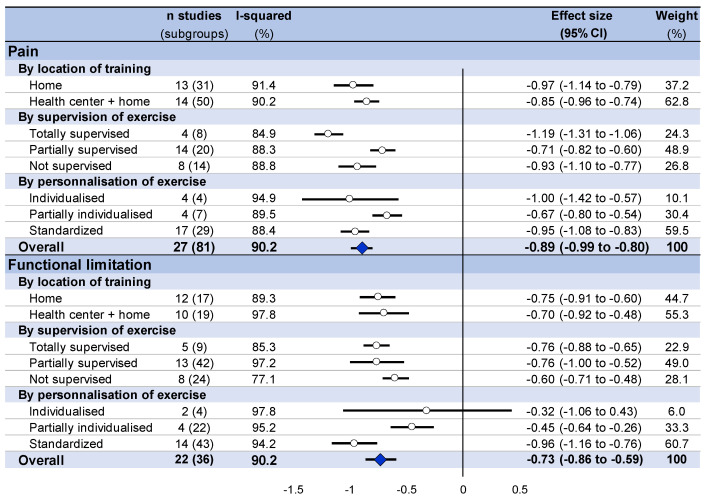
Summary of meta-analysis on the effect of home-based exercise on pain intensity and functional limitation, stratified by setting (exclusive home-based training versus home-based and other setting), supervision, and standardization of training.

**Figure 4 ijerph-18-08430-f004:**
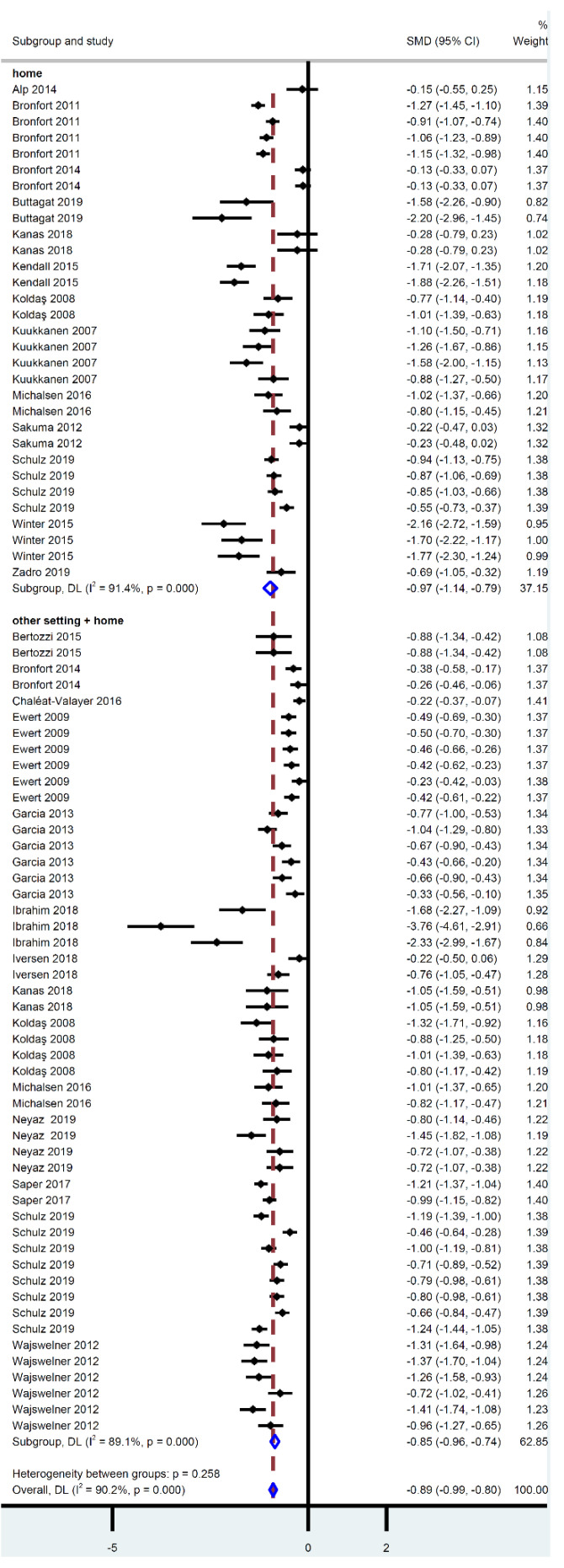
Meta-analysis on the effect of home-based exercise on pain intensity, stratified by setting (exclusive home-based training versus home-based and other setting).

**Figure 5 ijerph-18-08430-f005:**
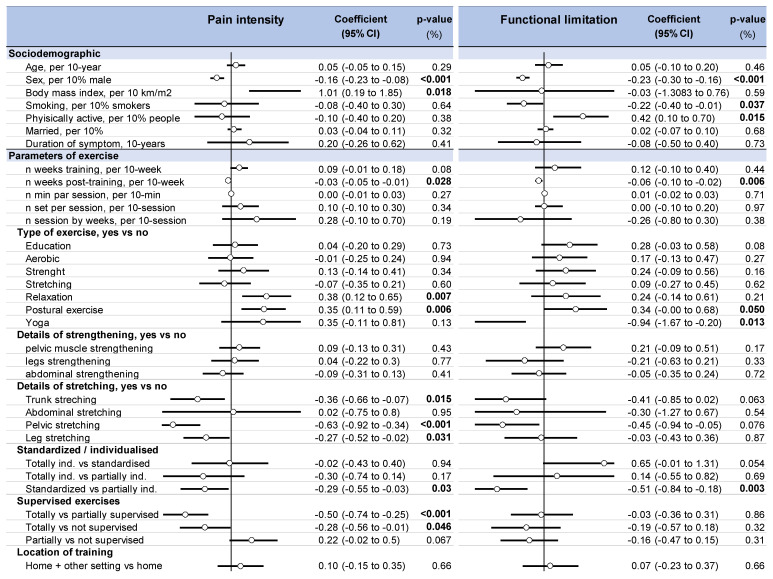
Metaregressions i.e., putative influencing variables on pain intensity and functional limitation following home-based exercise in LBP.

**Table 1 ijerph-18-08430-t001:** Characteristics of included studies.

	*Study*		*Characteristics of Indivuals*			*Exercise Setting*		*Type of Intervention*								*Volume of*	*Exercise*		*Outco*	*Mes*
*Study*	Design	Country	n gp	Age (Mean ± SD)	Sex (% Men)	Home	Other	Strength	Stretching	Relaxation	Aerobic	Education	Postural	Yoga	Other	Number of weeks	n Session/Week	Duration of Session	Pain	Functionnal Disability
*Alp 2014*	RCT	Turkey	24	48 ± 27.5	0	X		X	X							6 weeks	3	45–60 min	VAS	RMDQ
			24	51 ± 39.8	0		X	X	X								7	?		
*Ben Salah Frih 2009*	RCT	Tunisia	54	34.7 ± 1.14	24.1	X	X	X	X							4 weeks	7	30 min	VAS	
			53	36.9 ± 1.29	26.4		X	X	X						Proprioception exercise	4 weeks	3	90 min		
*Bernadelli 2020*	RCT	Italy	51	50.5 ± 9.7	25.5		X	X	X			X				7 weeks	1	30 min		RMDQ
			50	51.9 ± 8.1	16	X			X			X								
*Bertozzi 2015*	RCT	Italy	20	42.7 ± 8.7	-	X	X		X	X			X			5 weeks	2	60 min	VAS	RMDQ
			20	47.5 ± 7.5	-	X			X	X			X				?	?		
*Bronfort 2014*	RCT	USA	96	57.1 ± 12	41	X	X	X				X	X			12 weeks	7	?	NPRS	RMDQ
			96	57.7 ± 11.9	32	X		X				X	X							
*Bronfort 2011*	RCT	USA	100	44.5 ± 11.8	43			X	X		X					12 weeks	2	60 min		
			100	45.2 ± 10.8	44										Spinal manipulative therapy	12 weeks	2	60 min	NPRS	RMDQ
			101	45.6 ± 10.3	41.6	X		X	X			X				12 weeks	7	60 min		
*Buttagat 2019*	RCT	Thailand	11	39.7 ± 17.4	27.2	X			X						Thai self-massage	4 weeks	3	?	VAS	ODI
			11	41.3 ± 15.8	9	X						X				?	?	?		
*Chaléat-Valayer 2016*	RCT	France	171	47.1 ± 8.5	23	X	X	X	X			X	X		rhythmic exercise	2 years	7	10 min	VAS	QBPDS
*Descarreaux 2002*	RCT	Canada	10	33.1	70	X		X								3 weeks	14	?	VAS	ODI
			20	35.0	35	X		X	X							6 weeks	14	?		
*Ewert 2009*	RCT	Germany	100	37.9 ± 11.6	8.0	X	X	X	X	X	X	X	X		cognitive– behavioral approach/stress control + warm up	13 weeks	2	60 min	WHYMPI	sf36-pcs
			102	41.1 ± 10.8	5.9	X	X	X	X	X	X				Warm up	13 weeks	?	?		
*Frost 1998*	RCT	England	36	34.2 ± 9.4	-	X	X		X	X	X	X	X			4 weeks	2	?		MODI
			35	38.5 ± 9.3	-	X				X		X	X				?	?		
*Garcia 2013*	RCT	Brasil	74	54.2 ± 1.57	31.0	X	X		X			X	X			24 weeks	7	240 min	VAS	RMDQ
			74	53.7 ± 1.53	21.6	X	X	X				X	X							
*Goode 2018*	RCT	England	20	69.6 ± 3.5	95	X		X	X	X	X	X								
			20	69.5 ± 4,0	90	X		X	X	X	X	X			Activity pacing + Cognitive restructuring	12 weeks	?	?		RMDQ
			20	71.9 ± 6.5	95	X			X											
*Groessl 2017*	RCT	USA	76	53.3 ± 12.7	73	X	X							X	Breathing	12 weeks	7	60 min	BPI	RMDQ
			76	53.6 ± 13.9	75											none	-	-		
*Haufe 2017*	RCT	Germany	112	43.5 ± 9.7	-	X	X	X								20 weeks	3	20 min		
			114	41.9 ± 10.6	-											20 weeks	3	20 min	VAS	ODI
			10	49.9 ± 8.8	80	X	X	X	X		X	X				6 weeks	2	20 min		
*Ibrahim 2018*	RCT	Nigeria	10	48.5 ± 14.9	70	X	X	X	X		X	X				6 weeks	2	20 min	NPRS	ODI
			10	50.3 ± 9.1	90	X	X		X		X	X					1	?		
*Iversen 2018*	RCT	Norway	37	43 ± 13	46	X	X	X	X		X	X				12 weeks	4	?		
			37	47 ± 11	41	X	X	X	X	X		X			ball games + body awareness + circle training	12 weeks	4.5	?	NPRS	ODI
			34	45 ± 15	55											none	-	-		
*Kanas 2018*	RCT	Brasil	17	-	-	X		X	X		X					8 weeks	3	?	NPRS	RMDQ
			13	-	-	X	X	X	X		X									
*Kendall 2015*	RCT	Canada	40	33	55	X		X			X	X			RTUS	6 weeks	?	?	VAS	ODI
		Australia	40	41	40	X		X			X	X			RTUS					
*Koldaş 2008*	RCT	Turkey	20	37.1 ± 6.5	21.1	X	X	X	X		X	X					10	?		
			20	41.5 ± 8.3	22.2	X	X	X	X			X				6 weeks	7	?	VAS	RMDQ
			20	42.1 ± 9.5	22.2	X		X	X			X					7	?		
*Kuukkanen 2000*	RCT	Finland	29	39.9 ± 8.9	-		X	X							warm-up + Balance + coordination	12 weeks	?	?		
			29	39.9 ± 7.9	-	X		X								12 weeks	?	?		ODI
			28	39.9 ± 7.9	-											none	-	-		
*Kuukkanen 2007*	RCT	Finland	28	40 ± 7.9	46.4	X		X	X							12 weeks	7	?	Borg CR-10 scale	ODI
*Miller 2007*	RCT	England	98	44.1 ± 16.2	-	X										5 weeks	?	?		
			137	43.7 ± 14.8	-	X										5 weeks	?	?		RMDQ
			150	44.9 ± 15.4	-											-	-	-		
*Michalsen 2016*	RCT	Germany	32	55.5 ± 10.6	-	X	X								Jyoti meditaion	8 weeks	7	25 min	VAS	RMDQ
		Germany	36	54.8 ± 10.6	-	X		X	X									20 min		
*Neyaz 2019*	RCT	India	35	33	-	X	X	X	X	X		X				6 weeks	7	30–35 min	DVPRS	RMDQ
			35	38	-	X	X		X	X										
*Shirado 2010*	RCT	Japan	103	42.0 ± 11.6	47.5	X		X	X			X				8 weeks	14	?	VAS	
			98	42.5 ± 12.3	40.8	X									Massage		?			
*Schulz 2019*	RCT	USA	81	72.5 ± 5.6	43.2	X	X	X	X		X	X			Balance exercise + massage		2	45–60 min		
			80	73.6 ± 5.3	52.5	X	X	X	X		X	X			Balance exercise + massage	12 weeks	1	60 min	NPRS	RMDQ
			80	74.7 ± 5.6	50.0	X		X	X		X	X					1	15 min		
*Saper 2017*	RCT	USA	127	46.4 ± 10.4	-	X	X			X				X	breathing		?	75 min		
			129	46.4 ± 11,0	-	X	X				X	X				12 weeks	7	60 min	NPRS	RMDQ
			64	44.2 ± 10.8	-							X					?	?		
*Sakuma 2012*	RCT	Japan	67	32.6 ± 11.5	-	X				X				X		2 weeks	7	9 min	VAS	
			31	35.8 ± 13												none	-	-		
*Tottoli 2019*	RCT	Brazil	72	-			X	X	X				X			6 weeks	2	50 min	VAS	BQBPSDQ
			72			X		X	X				X		Warm up					
*Wajswelner 2012*	RCT	Australia	44	49.3 ± 14.1		X	X						X		Breathing	24 weeks		60 min	NPRS	
			43	48.9 ± 16.4		X	X	X	X						Swiss ball					
*Winter 2015*	RCT	Australia	12	45.9 ± 13.3		X			X								5			
			13	48.9 ± 7.2		X			X							6 weeks	5		NPRS	MODI
			13	38.3 ± 12.8		X		X	X								3			
*Zadro 2019*	RCT	Australia	30	68.8 ± 5.5	40.0	X		X	X		X					8 weeks	3	60 min	NPRS	RMDQ
			30	67.8 ± 6	56.6											none				

## Data Availability

All available data are included in this article.

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
