# Peer review of "Effect of Home Exercise Training in Patients with Nonspecific Low-Back Pain: A Systematic Review and Meta-Analysis"

_ijerph, 2021, doi:10.3390/ijerph18168430_

Round 1

Reviewer 1 Report

The authors are presenting the interesting theme of the efficacy of exercise on low back pain through systematic review. 
Points need to be improved start from the concept itself. The authors demonstrated that they included 'non-specific LBP' but did not clarify the evidence selection process and detailed inclusion and exclusion criteria regarding how to assess and select the articles appropriately. Need to clarify the process with the flow chart. What is the number of Science Direct as 'n=23 690'? How could the authors deduct the final article list? 

What was the definition of 'non-specific low back pain' and 'home exercise' from the conceptualization of the research questions? Spinal manipulative therapy is 'home exercise'? Need to clarify the definition of the terminology and the development of the search strategy with the clear inclusion & exclusion criteria of the disease & interventions entities.

Author Response

Reviewer 1

The authors are presenting the interesting theme of the efficacy of exercise on low back pain through systematic review.

[REPLY] Thank you for your positive comment

Points need to be improved start from the concept itself. The authors demonstrated that they included 'non-specific LBP' but did not clarify the evidence selection process and detailed inclusion and exclusion criteria regarding how to assess and select the articles appropriately. Need to clarify the process with the flow chart. What is the number of Science Direct as 'n=23 690'? How could the authors deduct the final article list?

[REPLY] Thank you for your positive comment. We added the definition of non-specific LBP in the Methods – search strategy section, and we clarify the inclusion and exclusion criteria. The Methods – search strategy section now reads: “We reviewed all studies reporting the effect of home-based exercise training on nonspecific LBP i.e. LBP not consecutive to a specific pathology such as inflammatory, tumoral or infectious process [3]. Animal studies were excluded. The PubMed, Cochrane Library, Embase and ScienceDirect databases were searched until April 20th, 2021, with the following keywords: low back pain AND (exercise OR physical) AND home (details for the search strategy used within each database are available in Appendix 1). The search was not limited to specific years and no language restrictions were applied. To be included, articles had to meet simultaneously the five following inclusion criteria: (1) randomized controlled trials (RCTs), (2) population ≥ 16 years old, (3) with non-specific LBP (chronic or not), (4) evaluation of at least one of our main clinically relevant out-come i.e. pain intensity or functional limitation, and (5) studies including home-based exercise therapy. […] We excluded studies assessing patients with specific LBP – i.e. caused by a specific cause such as pregnancy or pathological entities. Conference, congress, or seminars, were excluded. In addition, reference lists from all publications meeting the inclusion criteria were manually searched to identify any further studies not found through the electronic search. Ancestry searches were also completed on previous re-views to locate other potentially eligible primary studies. Two authors (Chloé Quentin and Reza Bagheri) conducted the literature searches, reviewed the abstracts, and based on the selection criteria, decided the suitability of the articles for inclusion, and extracted the data. When necessary, disagreements were solved with a third author (Frédéric Dutheil).” The flow chart (Figure 1) now matches the selection process. As soon as an article did not meet one of our inclusion criteria, the article was not included. The number 'n=23 690' for ScienceDirect is the number of articles available using the keywords. In fact, a meta-analysis is time-demanding because of the search strategy and the selection of articles to be included. Please see Appendix 1. The final article list is composed of articles meeting the requirement of the search strategy i.e. all inclusion and exclusion criteria.

What was the definition of 'non-specific low back pain' and 'home exercise' from the conceptualization of the research questions? Spinal manipulative therapy is 'home exercise'? Need to clarify the definition of the terminology and the development of the search strategy with the clear inclusion & exclusion criteria of the disease & interventions entities.

[REPLY] Thank you for your positive comment. The definition of non-specific LBP appears now three times, once in the introduction i.e. “Non-specific LBP is defined as a LBP not attributed to recognisable known specific pa-thology (e.g. inflammatory, tumoral or infectious process) [3].”, and twice in the search strategy “non-specific LBP i.e. LBP not consecutive to a specific pathology such as inflammatory, tumoral or infectious process [3]” and “We excluded studies assessing patients with specific LBP – i.e. caused by a specific cause such as pregnancy or pathological entities.” We now specify the definition of home exercise in the Methods section i.e. “Home exercise programs are series of exercises that patients complete at home for therapeutic gains or physical capacity – home exercises are designed to be practical, accessible and feasible so that patients can maximize efforts.” Therefore, spinal manipulative therapy is not considered as home exercise.

Reviewer 2 Report

Conclusions:

• The paper is a valuable review;
• The language use in the paper is here and there poor and/or erroneous. Below I suggest deleting wrong expressions, or improving sentences;
• Figure 1 is incomplete. Between the 24 270 papers and the selected 31 is information missing. You should here explain some of the work reported in Appendix 1. For example, it might be better to merge (part of) the queries in Appendix 1 in Figure 1 at lines 156-157. If you do not like that, then please refer to Appendix 1 in Figure 1, because the info is missing in the figure. You find details about this below in points 6. and 7;
• The conclusion is not crisp about home exercise training. From the point of view of a municipality officer, the conclusion should be made sharper. For instance: “From the literature, it is concluded that home training successful, if the training is done at home with friends from a community group taking part, etc … “. Would the conclusion become improved if Appendix 2 is fully employed in the text? Apepndix 2 holds a lot of information!

Observations and remarks about the text, to repair the paper:

1. line 61, “… some meta-analysis have suggested …” should read plural “… some meta-analyses have suggested …”;
2. line 61-64, Has a wrong reference [11]. Owen’s paper [11] is on the best way to transfer clients from wheelchair to toilet and vica versa. Check: https://pubmed.ncbi.nlm.nih.gov/1824751/ ;
3. line 64, “European recommendation highlighted …” could better read in the context here “But European recommendation highlighted …”;
4. line 66, “… improve the best outcomes in exercise center …” should read “… improve the best outcomes in exercise centers …” or should read “… improve the best outcomes in an exercise center …”;
5. line 66-67, “… but similarly, data are also lacking for home-based exercise, above formula …” seems redundant: you mentioned in line 65 already that data are scarce! Is the term “similarly” incorrect here? In French you say (translated by Google): “Les exercices à domicile promettent de faire face à la disponibilité des centres d'exercice.” Is this what you want to say? If not please edit your English expression;
6. line 168-169, the sum is wrong: it should read 24 699, according to the sum in lines 161-163;
7. line 171-177, seem to be misplaced. Please correct Figure 1 where needed. The count of the excluded papers sums to 527 = 133+50+163+ … +9. To me it seems that the misplaced counts should be put in the Fig. 1 between lines 163 and 167. The numbers in lines 161-163 sum up to 24 699. Then subtract the exclusion counts: 24 699 - 527 = 24 172;
8. lines 180-181, are peculiar. I do not understand what Keywords led you to exclude over 24 000 studies? Is it a keyword not mentioned in the heading lines 156-157 of Figure 1? For instance the Keyword “Home”?
9. line 559, “… had only groups exercising at home.” could be said clearly “… had groups exercising only at home.”;
10. line 567, “… size of 10 000 individuals …” is it “… size of over 10 000 individuals …”?
11. line 569, “… had several measurement times, …” needs clarification. Do you mean: “… had several measurement interval times, …”? Or do you mean training duration?
12. line 578, this chapter should be deleted;
13. line 581-589, are bad and partially conflicting & illogical order. This has to be rewritten;
14. line 590, is wrong. Remove the redundant words and apostrophes.
15. line 591-592, 595, remove redundant apostrophes;
16. line 677-678, you register here 4 groups, while in the Koldas paper [36] and in Table 1 you only have 3 groups. What in the lines 677-678 should be deleted?
17. line 816-818, is wrong. The Cochrane reference gives at July 6, 2021: https://www.cochranelibrary.com/doi/10.1002/14651858.CD011284.pub2 does not exist anymore. It says “Error 404, The page you requested does not appear to be here. It might have been changed, removed, or might be temporarily unavailable.”
18. line 830, this reference is not completed;
19. line 831, is wrong

Author Response

The paper is a valuable review

[REPLY] Thank you for your positive comment

The language use in the paper is here and there poor and/or erroneous. Below I suggest deleting wrong expressions, or improving sentences

[REPLY] Thank you for improving the readability of our article.

Figure 1 is incomplete. Between the 24 270 papers and the selected 31 is information missing. You should here explain some of the work reported in Appendix 1. For example, it might be better to merge (part of) the queries in Appendix 1 in Figure 1 at lines 156-157. If you do not like that, then please refer to Appendix 1 in Figure 1, because the info is missing in the figure. You find details about this below in points 6. and 7;

[REPLY] Thank you for your positive comment. We updated the Figure 1 so the search strategy now matches the selection process.

The conclusion is not crisp about home exercise training. From the point of view of a municipality officer, the conclusion should be made sharper. For instance: “From the literature, it is concluded that home training successful, if the training is done at home with friends from a community group taking part, etc … “. Would the conclusion become improved if Appendix 2 is fully employed in the text? Appendix 2 holds a lot of information!

[REPLY] Thank you for your positive comment. We added your suggestion at the beginning of the conclusion. Appendix 2 is now Figure 3. We also updated Figure 2 (0.85 instead of 0.89 for Pain / by location of training / Health center + home.

Observations and remarks about the text, to repair the paper:

1.line 61, “… some meta-analysis have suggested …” should read plural “… some meta-analyses have suggested …”;

[REPLY] Thank you for your positive comment. We corrected that mistake. 11.   Owen, P.J.; Miller, C.T.; Mundell, N.L.; Verswijveren, S.J.J.M.; Tagliaferri, S.D.; Brisby, H.; Bowe, S.J.; Belavy, D.L. Which Spe-cific Modes of Exercise Training Are Most Effective for Treating Low Back Pain? Network Meta-Analysis. Br J Sports Med 2020, 54, 1279–1287, doi:10.1136/bjsports-2019-100886

  1. line 61-64, Has a wrong reference [11]. Owen’s paper [11] is on the best way to transfer clients from wheelchair to toilet and vica versa. Check: https://pubmed.ncbi.nlm.nih.gov/1824751/

[REPLY] Thank you for your positive comment. We corrected that mistake.

  1. Owen, P.J.; Miller, C.T.; Mundell, N.L.; Verswijveren, S.J.J.M.; Tagliaferri, S.D.; Brisby, H.; Bowe, S.J.; Belavy, D.L. Which Spe-cific Modes of Exercise Training Are Most Effective for Treating Low Back Pain? Network Meta-Analysis. Br J Sports Med 2020, 54, 1279–1287, doi:10.1136/bjsports-2019-100886

  1. line 64, “European recommendation highlighted …” could better read in the context here “But European recommendation highlighted …”;

[REPLY] Thank you for your positive comment. We corrected that mistake.

  1. line 66, “… improve the best outcomes in exercise center …” should read “… improve the best outcomes in exercise centers …” or should read “… improve the best outcomes in an exercise center …”;

[REPLY] Thank you for your positive comment. We corrected that mistake.

  1. line 66-67, “… but similarly, data are also lacking for home-based exercise, above formula …” seems redundant: you mentioned in line 65 already that data are scarce! Is the term “similarly” incorrect here? In French you say (translated by Google): “Les exercices à domicile promettent de faire face à la disponibilité des centres d'exercice.” Is this what you want to say? If not please edit your English expression;

[REPLY] Thank you for your positive comment. The sentence now reads: For home-based exercise in LBP, data are scarce. Although data are lacking for home-based exercise, as well as for the optimal intensity, frequency and duration of exercise, supervised exercise seemed to improve the best outcomes in exercise centers [2,5].

  1. line 168-169, the sum is wrong: it should read 24 699, according to the sum in lines 161-163;

[REPLY] Thank you for your positive comment. We corrected that mistake.

  1. line 171-177, seem to be misplaced. Please correct Figure 1 where needed. The count of the excluded papers sums to 527 = 133+50+163+ … +9. To me it seems that the misplaced counts should be put in the Fig. 1 between lines 163 and 167. The numbers in lines 161-163 sum up to 24 699. Then subtract the exclusion counts: 24 699 - 527 = 24 172;

[REPLY] Thank you for your relevant comment. The flow chart (Figure 1) now matches the selection process. As soon as an article did not meet one of our inclusion criteria, the article was not included.

  1. lines 180-181, are peculiar. I do not understand what Keywords led you to exclude over 24 000 studies? Is it a keyword not mentioned in the heading lines 156-157 of Figure 1? For instance the Keyword “Home”?

[REPLY] Thank you for your relevant comment. The flow chart (Figure 1) now matches the selection process. As soon as an article did not meet one of our inclusion criteria, the article was not included.

  1. line 559, “… had only groups exercising at home.” could be said clearly “… had groups exercising only at home.”;

[REPLY] Thank you for your positive comment. We corrected that mistake.

  1. line 567, “… size of 10 000 individuals …” is it “… size of over 10 000 individuals …”?

[REPLY] Thank you for your positive comment. We corrected that mistake.

  1. line 569, “… had several measurement times, …” needs clarification. Do you mean: “… had several measurement interval times, …”? Or do you mean training duration?

[REPLY] Thank you for your positive comment. We mean both. We added your suggestions.

  1. line 578, this chapter should be deleted;

[REPLY] Thank you for your positive comment. We deleted it.

  1. line 581-589, are bad and partially conflicting & illogical order. This has to be rewritten;

[REPLY] Thank you for your positive comment. This part now reads: Author Contributions: “Conceptualization, C.Q. and F.D.; methodology, J.B.B.M. and F.D.; soft-ware and analysis, F.D. validation, all authors; writing—original draft preparation, C.Q. and F.D.; writing—review and editing, all authors.; supervision, F.D.;funding acquisition, N/A. All authors have read and agreed to the published version of the manuscript.”

  1. line 590, is wrong. Remove the redundant words and apostrophes.

[REPLY] Thank you for your positive comment. We deleted it.

  1. line 591-592, 595, remove redundant apostrophes;

[REPLY] Thank you for your positive comment. We deleted it.

  1. line 677-678, you register here 4 groups, while in the Koldas paper [36] and in Table 1 you only have 3 groups. What in the lines 677-678 should be deleted?

[REPLY] Thank you for your attentive reading. The study from Koldas et al. has three groups as mentioned in Table 1. However, only two groups have home-exercise, thus only those two groups were included within our article. Line 677-678 refers to Appendix 2 that became Figure 3 as you suggested. As stated in the Results 3.10 Senstivity analyses section: “We also computed sensitivity analyses using only the median time of follow-up i.e. three months (Appendix 4 for meta-analysis and Appendix 5 for metaregressions), and also using the last time of follow-up (Appendix 6 and 7).” i.e. Figure 3 (ex Appendix 2) used all measurement time. As there are two measurement time per group, we do have 2 measurement time x 2 groups = 4 lines for Koldas et al. In order to clarify the understanding for readers, we added the following sentence in the Methods – Statistics section: “We computed aforementioned meta-analysis using all measurement time. To verify the strength of our results, we computed sensitivity analyses using only the median time of follow-up and then using only the last time of follow-up.”

  1. line 816-818, is wrong. The Cochrane reference gives at July 6, 2021: https://www.cochranelibrary.com/doi/10.1002/14651858.CD011284.pub2 does not exist anymore. It says “Error 404, The page you requested does not appear to be here. It might have been changed, removed, or might be temporarily unavailable.”

[REPLY] Thank you for your positive comment. We fixed it.

  1. line 830, this reference is not completed;

[REPLY] Thank you for your positive comment. We fixed it.

  1. line 831, is wrong

[REPLY] Thank you for your positive comment. We fixed it.
